# Rodent Models of Alzheimer’s Disease: Past Misconceptions and Future Prospects

**DOI:** 10.3390/ijms25116222

**Published:** 2024-06-05

**Authors:** Helen M. Collins, Susan Greenfield

**Affiliations:** Neuro-Bio Ltd., Building F5 The Culham Campus, Abingdon OX14 3DB, UK; susan.greenfield@neuro-bio.com

**Keywords:** Alzheimer’s disease, Parkinson’s disease, animal models, mice, rats, isodendritic core

## Abstract

Alzheimer’s disease (AD) is a progressive neurodegenerative disease with no effective treatments, not least due to the lack of authentic animal models. Typically, rodent models recapitulate the effects but not causes of AD, such as cholinergic neuron loss: lesioning of cholinergic neurons mimics the cognitive decline reminiscent of AD but not its neuropathology. Alternative models rely on the overexpression of genes associated with familial AD, such as amyloid precursor protein, or have genetically amplified expression of mutant tau. Yet transgenic rodent models poorly replicate the neuropathogenesis and protein overexpression patterns of sporadic AD. Seeding rodents with amyloid or tau facilitates the formation of these pathologies but cannot account for their initial accumulation. Intracerebral infusion of proinflammatory agents offer an alternative model, but these fail to replicate the cause of AD. A novel model is therefore needed, perhaps similar to those used for Parkinson’s disease, namely adult wildtype rodents with neuron-specific (dopaminergic) lesions within the same vulnerable brainstem nuclei, ‘the isodendritic core’, which are the first to degenerate in AD. Site-selective targeting of these nuclei in adult rodents may recapitulate the initial neurodegenerative processes in AD to faithfully mimic its pathogenesis and progression, ultimately leading to presymptomatic biomarkers and preventative therapies.

## 1. Introduction

Animal models are vital for furthering our understanding of disease mechanisms and for prompting new therapeutic strategies. A field with a clear need for a valid and translatable animal model is neurodegeneration, particularly Alzheimer’s disease (AD), which constitutes 60–70% of all dementia cases worldwide and is characterised by progressive and catastrophic neuronal degeneration, leading to cognitive and memory deficits [1]. Despite its prevalence, there are no disease-modifying treatments for AD. Current medications include donepezil, galantamine and memantine, which increase cholinergic synaptic transmission to ameliorate the cognitive deficits in mild AD, but do not improve the disease’s prognosis [2]. Emerging therapies, notably anti-amyloid beta (Aβ) monoclonal antibodies such as Lecanemab, may slow disease progression, but have also not been shown to cure AD [3,4]. There are also no biomarkers for the presymptomatic detection of AD. This deficiency is due to a poor understanding of the neuropathological mechanisms of AD, in turn resulting from a lack of authentic animal models. To understand the pathogenesis of AD and how it progresses through the brain, we need an animal model that reproduces the causal mechanism driving the disease. This review will outline the various apparent causes of AD and evaluate the success or otherwise of their reproduction in rodents. We shall then consider new insights into the pathological mechanisms of neurodegeneration with the potential for alternative, and likely more valid, rodent models.

## 2. Cholinergic Hypothesis

An early theory of the origin of cognitive impairments in AD was the loss of cholinergic signalling [5,6,7]. Post mortem studies show that AD brains have 75% fewer cholinergic neurons in the basal forebrain compared to healthy controls [8,9,10]. These cells also have significantly lower concentrations of acetylcholine (ACh) and its synthesising enzyme, choline acetyltransferase, in the hippocampus and cortex [5,11,12,13,14]. Given ACh plays a key role in cognition and memory [15], loss of cholinergic signalling may account for the memory deficits seen in mild to moderate AD [16,17]. To compensate for the loss of cholinergic neurons, drugs that enhance cholinergic transmission were developed as treatments for mild cognitive impairment (see [18]); the most common have been acetylcholinesterase (AChE) inhibitors, starting with tacrine in 1989 and now superseded by drugs such as Aricept (donepezil) and galantamine [19], which prolong the availability of ACh by impeding its metabolism.

Several approaches have been used to model cholinergic neuron loss in AD, typically in rats. The muscarinic receptor antagonist scopolamine, for example, induces memory deficits when injected into the rat basal forebrain [20,21]. Specific cholinergic neuron degeneration can also be produced by adding carrier molecules to non-specific toxins, such as 192 IgG-saporin, which result in degeneration of cholinergic cell bodies and terminals via apoptotic cell death [22,23,24]. Intraventricular infusion of 192 IgG-saporin into rats causes working memory impairments on the Morris water maze, a gold-standard test for spatial memory in rodents (behavioural deficits seen in rats with cholinergic lesions, see [23]). Alternative approaches to the selective destruction of cholinergic neurons include mechanical lesions to cholinergic nuclei in the brainstem, brain trauma and hyperthermia after post-ischemic brain injury [25].

While cholinergic models successfully identified AChE inhibitors as cognitive enhancers for the early stages of AD, their ability to model the disease in its entirety is questionable. For instance, the effects of neurotoxic lesions on behaviour are highly variable and not always detected [26,27]. Extensive cholinergic neuron loss, potentially as great as 75%, is needed to detect behavioural effects in rodents [28]. In contrast, post mortem imaging shows that even patients with severe AD (Braak stages IV-VI) only lost approximately one third of the volume of the nucleus basalis of Meynert (NbM), a midbrain cholinergic nucleus that is particularly susceptible to degeneration in AD [29]. The critical threshold for neuronal loss to become apparent as cognitive deficits [30] may therefore be much lower in humans than in these rodent models, meaning they do not replicate the initial stages of neurodegeneration. Furthermore, most experimental models use neurotoxins infused via the ventricles, meaning cell loss is brain-wide and not targeted to the cholinergic nuclei [23]. These models also do not mimic the classic molecular hallmarks of AD: amyloid plaques and neurofibrillary tau tangles (NFTs). If anything, these models demonstrate the importance of cholinergic transmission in cognition, but poorly recapitulate the disease phenotype in humans. Finally, not all cholinergic cells degenerate in AD, and not all cells that degenerate are cholinergic—for example, cholinergic interneurons in the striatum are spared even in patients with AD [31]. This double dissociation between AD and cholinergic transmission suggests that the latter is not the primary mechanism mediating pathogenesis of AD.

## 3. Amyloid Hypothesis

The ‘amyloid hypothesis’, or the ‘amyloid cascade hypothesis’, was first proposed in 1991 following the discovery of a mutation in amyloid precursor protein (APP) in the post mortem brains of patients with AD [32,33]. This theory proposes that the accumulation of a toxic form of Aβ, through abnormal cleavage of the precursor molecule APP [34] and/or reduced clearance [35], leads to the downstream aggregation of tau and cell death, thus precipitating neurodegeneration in AD. This hypothesis was supported by genetic studies that identified several causative familial AD mutations in the amyloid production pathway, including to APP, presenilin 1 (PS1) and presenilin 2 (PS2) [36,37,38]. Common human polymorphisms in the apolipoprotein E (APOE) gene, namely APOE ε4, also elevate Aβ levels and increase the risk of sporadic AD [39,40]. Increased plasma concentrations of Aβ correlate with increased risk of AD, with levels elevated years before the onset of symptoms [41]. Indeed, the presence of Aβ plaques is a criterion for a neuropathologic diagnosis of AD [42].

### 3.1. Amyloid Transgenic Rodents

The most common approach to reproduce amyloid dysregulation in AD is transgenic mice. Mice do not spontaneously develop amyloid pathology [43,44]: hence, transgenic mice express human genes associated with familial AD mutations, namely in APP and PS1 [45], under the assumption that the same pathways are involved in sporadic AD. The first successful AD transgenic mouse was developed by Games et al., (1995): researchers genetically modified mice to overexpress platelet-derived APP (PDAPP) with a human mutation. These mice exhibit increased Aβ and plaque formation, as well as cognitive and memory impairment with age [46,47]. Other APP-targeting mice models express two mutations of the same type (e.g., Tg2576; Table 1) or two different mutations (e.g., J20, PDGF-APPSw, Ind; Table 1). Bi-genic mice, such as the PSAPP strain, express mutations to both APP and PS1 (APP/PS1), whilst the most extensive amyloid model to date is the 5xFAD mouse, which contains five familial AD mutations (Table 1; [48]). Other strains combine mutations to other AD-associated pathways, for example overexpressing human APOE ε4 [18,39,49,50,51] or knocking out β-secretase 1 [52,53,54].

Most studies of amyloid-overexpressing mice show at least some degree of Aβ accumulation and deposition, but the various mutations produce different phenotypes. In general, the more mutations the strain contains, or the more genes targeted, the more aggressive and earlier are the AD-like neuropathology and behavioural impairments. Interestingly, pathology also appears to be more pronounced in female than in male mice [55]. Table 1 shows examples of the different amyloid transgenics and their phenotypes.

**Table 1 ijms-25-06222-t001:** Summary of amyloid transgenic rodents. Aβ, amyloid beta; APP, amyloid precursor protein; LC, locus coeruleus; MWM, Morris Water Maze; NOR, novel object recognition; PSEN1, presenilin 1.

Model	Modification	Pathology	Behaviour	References
Mice				
Single APP mutations e.g., PDAPP, APP(V717L), APP23 and APP E693Δ.	Overexpression of human mutations to APP (e.g., Swedish K670N/M671L, London V717L and Indiana V717F).	Elevated Aβ levels compared to controls. Age-dependent amyloid plaque deposition. Gliosis and inflammation. Loss of plasticity and dendrites in the hippocampus.	Spatial working memory deficits compared to wildtype controls (MWM, radial maze). Recognition memory deficits with age.	[46,56,57,58,59]
Double APP mutations (same mutation), e.g., Tg2576 (double Swedish).	Overexpression of two of the same mutations, e.g., Tg2567 have double Swedish mutation.	Elevated Aβ and amyloid plaques. Dendritic spine loss and synaptic transmission deficits. No evidence of neuronal loss. Exacerbated pathology compared to single mutants.	Impaired spatial and working memory form as early as 6 months old.	[60,61,62]
Double APP mutations (different mutations), e.g., J20, A7 APP, PDGF-APPSw, Ind and TgCRND8.	Overexpression of two different human APP mutations (e.g., J20 have Swedish and Indiana).	Elevated Aβ and amyloid plaques. Limited neuronal loss in the hippocampus. Neuroinflammation. Deficits in synaptic plasticity. Typically, more extensive than single mutants.	Impaired spatial and working memory form as early as 6 months old.	[62,63,64,65]
Single PS1 mutations e.g., PS1(A246E), PS1(M146L), PS1(M146V).	Overexpression of mutations to the human PSEN1 gene.	Elevated PSEN1 expression but no evidence of pathology up to 2 years old. Used mainly for the generation of APP/PS1 bi-genic strains.	No learning or spatial memory deficits.	[66,67,68,69,70]
APP/PS1 bi-genic strains, e.g., PSAPP, PS/APP, APP23 x PS1-R278I, PS/APP (Tg2576 x PS1 M146L) and APP(V717I) x PS1 (A246E).	Crossed APP-mutant and PS1 mutant mice.	Increased ratio of Aβ42 to Aβ40. Earlier amyloid plaque deposition than in single mutant strains. Accelerated and more extensive amyloidosis. Tau hyperphosphorylations but NFTs. Neuroinflammation from 8 months.	No MWM deficits at 9 months, but deterioration from 15–17 months. Deficits in spatial working memory on the Y maze from 3 months. Increased anxiety and depression-like behaviours, reduced sociability.	[41,69,71,72,73,74,75,76,77,78,79]
Triple mutant strains e.g., 3xTg-AD.	Mice overexpressing APP, PSEN1 and tau mutations.	Intracellular Aβ accumulation as early as 3 months old, plaques from 6 months. Tau aggregation from 12–15 months. Synaptic dysfunction. Gliosis and reactive microglia from 7 months old.	Memory retrieval deficits from 4 months, prior to plaques and NFTs. Deficits in spatial learning from 6 months.	[80,81,82]
5xFAD.	5xFAD on B6SJL or C57BL6 backgrounds (APP Swedish, Florida I716V and London; PSEN1 M146L and L286V).	Rapid cerebral accumulation of Aβ from 1.5 months. Reduced synaptic markers and cortical neuron loss. Gliosis and neuroinflammation.	Spatial memory impairment, working memory deficits, motor deficits. Age-dependent reduction in anxiety.	[48,83]
APP Knock-in e.g., APP^NL-F^ and APP^NL-G-F^.	Crispr/Cas9 used to humanise Aβ sequence in endogenous APP gene and add human mutations.	Humanised APP expression. Increased expression of Aβ isoforms.	Working memory, object recognition and social recognition memory deficits. Increased anxiety and depression-like behaviours. Reduced sociability.	[84,85,86,87,88]
Rats				
Double APP mutations, e.g., APP21, McGill-R-Thy1-APP.	Viral vectors introduced human mutations to zygotes containing two different human APP mutations, e.g., McGill-R-Thy1-APP have Swedish and Indiana.	Aβ pathology from 7 days old. Plaques from 6 months old, brain-wide by 18 months old. Age-dependent gliosis, synapse loss and transmission deficits.	Poor spatial memory acquisition on the MWM, similar object recognition memory to controls. Deficits in fear conditioning.	[89,90,91]
APP/PS1 bi-genic, e.g., APP + PS1, Tg478, Tg1116 Tg344-AD (APP Swedish, PS1 ΔE9).	Viral vectors introduced human mutations to zygotes containing mutations to APP and PS1.	Aβ42 levels greater and plaques appeared earlier than in APP-only mutants, age-dependent increases in expression and distribution. Increased tau expression, including in tangle-like structures. Neuron loss in the hippocampus and LC.	Deficits in spatial memory from 6–10 months old. Deficits in recognition memory (NOR) from 24 months. Some evidence of increased anxiety in older animals.	[92,93,94,95,96,97,98]
APP knock-in strains.	Crispr/Cas9 used to humanise Aβ sequence in endogenous APP gene; others include human mutations to APP gene.	Humanisation of APP (wildtype human gene) increased Aβ40 levels but not Aβ42 or accumulation, no plaques or NFTs detected up to 2 years old. Humanised genes with mutations, e.g., KM670/671NL double Swedish mutation have increased Aβ42 and altered synaptic transmission, tau phosphorylation, gliosis and neuron loss (in young rats).	Mostly, behaviour is not distinguishable from that of wildtype controls. APP^NL-G-F^ rats have deficits on MWM from 5–7 months old, some evidence of increased anxiety.	[88,99,100,101]

Despite their extensive use, transgenic models have fundamental limitations for studying AD. Firstly, they are based on familial mutations, which only 5–10% of patients with familial AD express [38]. For patients with sporadic AD, there is little indication that disruption to these genes plays any role in the pathogenesis of the disease [102]. Moreover, unlike transgenic mice that can express up to five different mutations [42,48], there are strikingly few examples of patients having multiple AD-associated mutations, even in familial AD [103]. A single mutant transgenic may therefore model specific familial AD mutations, but this does not reflect the causal mechanisms of sporadic disease. 

Furthermore, the overexpression of mutant human genes in addition to the endogenous mouse genes leads to non-physiological levels of protein expression. Even the PDAPP mouse with just one mutation to APP has ten-fold more APP than wildtype controls [46]. This excessive protein expression is not representative of patients with familial AD mutations, let alone sporadic AD. While knock-in models humanise existing rodent genes to induce mutations and therefore have more physiological levels of Aβ (e.g., [84,104]), they are still often bred onto APP-transgenic lines to accelerate the appearance of AD-like phenotypes, and thus still rely on transgene overexpression (e.g., [84]).

While familial AD patients express mutations throughout life, AD-associated gene expression varies greatly across age and disease stage [105,106]. Particularly relevant to sporadic AD is that there may be changes in gene expression or activity of proteins that does not become apparent until adulthood. Transgenic rodents, on the other hand, will exhibit brain-wide changes in protein expression throughout development and in the postnatal period. This potentially alters normal brain development and therefore the progression of AD-like pathology in adulthood. A better model of AD would therefore trigger changes in gene expression or alter protein expression in adult animals, to recapitulate the age-dependent nature of the disease.

### 3.2. Seeded Amyloid Models

More recently, “seeded” models have been used to localise AD-like pathology in specific brain structures and to accelerate the onset and severity of amyloid deposition in adult animals [107]. This procedure involves injecting Aβ, whether that be synthetic oligomers or brain homogenates from transgenic mice or patients with AD, into the brains of transgenic mice. Typically, injections are targeted to the hippocampus and the overlaying cortex [107]; for example, bilateral hippocampal injections of Aβ into the of 5-month-old APP mice produced amyloid plaques in the hippocampus one month earlier than in sham-injected APP23 mice [108]. Seeding Aβ into the olfactory bulb in 2-month-old 5xFAD mice (before amyloid deposits are detectable) increased Aβ expression 12 weeks later compared to non-injected 5xFAD mice. This resulted in reduced neurogenesis and activity of the targeted cells, as well as olfactory function [109]. Seeding Aβ into the hippocampus of APP-mutant mice reduced hippocampal neurogenesis and proliferation, and increased cell death compared to controls [110].

Nonetheless, the effect of seeding on amyloid pathology is highly variable and depends on both the source of Aβ and the host. For instance, APP/PS1 mice injected with APP23 homogenates develop hippocampal plaques that are more diffuse than APP/PS1 injected with APP/PS1 homogenates [108,111]. These discrepancies make it difficult to compare results between strains and constructs. Moreover, it has not yet been possible to induce AD-like pathology by injecting mutant Aβ into wildtype mice, nor can non-transgenic homogenates produce amyloid pathology in young transgenic hosts [107,109]. This is likely because endogenous Aβ expression is too low to maintain seeded Aβ in wildtype animals, and aggregated Aβ is needed to further misfold endogenous Aβ in transgenics. Overall, these failures demonstrate that simply increasing Aβ levels in the rodent brain is not sufficient to trigger downstream AD pathology, thereby questioning the validity of amyloid seeding in modelling the causative mechanisms of AD.

### 3.3. Limitations of Amyloid-Targeting AD Models

Rodent models overexpressing amyloid are predicated on the assumption that the accumulation of Aβ in the brain is the cause of AD. Yet, it is becoming increasingly apparent that this peptide is not likely to be the primary agent driving neurodegeneration. For example, post mortem brains of elderly patients without dementia can have extensive amyloid deposits, suggesting that Aβ alone is not sufficient to produce AD [112]. Moreover, amyloid deposits are absent in the early stages of AD, yet other AD-associated pathology is already present, such as NFTs (see following section). Models based on amyloid overlook vital neuropathological changes that occur before Aβ accumulation, meaning they do not simulate the true cause of the disease. The lack of tau pathology in amyloid transgenics (e.g., [48]) in these models is perhaps their most important limitation for authentic modelling of AD.

Amyloid-targeting models also fail to replicate the onset of AD-like pathology. In transgenic mice, abnormalities appear much earlier than even in early-onset AD: for example, 5xFAD mice display increased Aβ levels at 1.5 months old [48]. In contrast, Aβ levels start to rise as early as twenty years before the onset of cognitive symptoms in patients [113]. Not only does this disparity suggest that these models are not completely authentic, but the accelerated emergence of Aβ pathology also restricts the window for the potential detection of presymptomatic markers. 

Despite these drawbacks, APP-transgenic mice have been central in the development of anti-Aβ therapies. For instance, immunisation of PDAPP mice with Aβ decreased amyloid pathology [114] and prevented cognitive deficits in other mouse strains [64,115,116]. Studies such as these have led to drugs such as Lecanemab and Donanemab (monoclonal antibodies against Aβ), which show some efficacy in reducing plaque burden in patients [3,4]. Nonetheless, Aβ-targeting drugs still routinely underperform in clinical trials in that they reduce amyloid burden but do not ameliorate cognitive deficits [117] and have not been shown to improve life expectancy [3,118]. The lack of translatability between encouraging results in rodents and failures in humans further suggests that amyloid-targeting animal models have poor predictive validity for AD. Other potential mechanisms should therefore be explored in models with the aim of producing more successful treatments in the future.

## 4. Tau Hypothesis

Tau is a microtubule-associated protein synthesised from the *MAPT* gene that stabilises microtubules [119]. In AD, tau becomes hyperphosphorylated and aggregates into NFTs [120], destabilising microtubules and thus leading to cell damage and eventual loss, although the exact mechanisms remain unclear [121]. While historically thought to be a consequence of Aβ accumulation [112], tau pathology can appear in the AD brain before, and independently of, Aβ deposits [122,123]. Moreover, the highly predictable spread of hyperphosphorylated tau correlates better with cognitive decline than amyloid pathology [124,125], suggesting that tau may be the initiating agent for neurodegeneration in AD [126].

### 4.1. Tau Transgenic Rodents

The induction of tauopathies in mice is difficult as endogenous murine tau does not aggregate into NFTs [44]. Thus, like amyloid-targeting transgenic models, human tau with one or more familial AD-associated mutations, predominantly P301L or P301S mutations, are instead overexpressed to induce high levels of hyperphosphorylated tau and consequently neurodegeneration [127,128,129,130,131,132]. Overexpression of mutant tau in h2N4P301L mice accordingly produced a four-fold increase in tau expression at 6–7 months old, which aggregated into NFTs. There was also evidence of tau aggregation and neuronal loss in the hippocampus and neocortex of hAPP-3RTau mice, an APP-mutant line with three additional mutations to tau [133]. This cellular phenotype was associated with motor and cognitive deficits, comparable to those seen in advanced AD [134]. Interestingly, hyperphosphorylated and aggregated tau has also been detected in mice expressing non-mutant human tau [135]. Human tau mutations also increase phosphorylated tau and produce NFTs when expressed in rats, although generally in the absence of significant neuronal loss (reviewed in [25]).

Nonetheless, the phenotypes of different tau transgenics vary depending on the human mutations they express and under which promoters. For instance, the P301L mutation under Thy-1 regulation produced a 0.7-fold increase in tau expression in 3-month-old mice, whereas under the CaMKII promoter, tau expression was 13-fold greater from 1 month old [130,134,136,137]. It is therefore difficult to generalise findings to other transgenics, let alone to the human condition.

The lack of amyloid plaques in tau-only mutants also challenges the authenticity of these models for AD. To overcome this issue, Oddo et al. developed the 3xTg-AD strain, which expresses human mutations to APP (Swedish), PS1 (M146V) and tau (MAPT P301L). This mouse exhibits brain-wide, age-dependent expression of amyloid plaques, tau hyperphosphorylation and progressive cognitive deficits [80,138,139]. Unlike other transgenic mice that exhibit hippocampal neurodegeneration from an early age [140,141], this strain shows limited neuronal loss in the hippocampus even up to 17 months old. This resembles the hippocampal degeneration seen in later stages of human AD (where hippocampal volume loss occurs after degeneration of other structures [123] and correlates with cognitive decline, i.e., is more severe in later stages of the disease [142,143]). Nonetheless, the concurrent presence of Aβ and tau throughout development and maturation in 3xTg-AD mice does not reflect the progressive and potentially sequential appearance of these markers in AD. The 3xTg-AD strain may therefore be a good model of late-stage AD pathology, but poorly recapitulates its early pathogenesis.

### 4.2. Seeded Tau Models

Tau can also be ‘seeded’ into the brain to induce neurodegenerative pathology. This process involves the stereotaxic injection of tau protein into the hippocampus and/or overlying cortex. For example, injecting synthetic preformed fibrils of tau into the hippocampus led to the spread of NFT-like inclusions to anatomically connected structures, including the thalamus and cortex [144,145]. Injections of human brain extracts from patients with high levels of tau also produced similar spread in P301L tau transgenic and 5xFAD mice [146,147], as well as in mice expressing wildtype human tau (without AD mutations; [148]). Although these seeding models, by definition, localise tau pathology to specific brain regions, the spread of aggregated tau to other structures differs from the clinical progression of AD [144,149]. Indeed, one study found that the transentorhinal cortex was spared from tau pathology after injection of synthetic preformed tau fibrils, yet this is the first cortical structure to show tau aggregation in human AD [123,149]. 

An alternative approach is to trigger the conditional and inducible overexpression of mutant tau genes in wildtype rodents via AAV-virus transfection. In one example, AAV-hTau-P301L injected into the cortex of wildtype mice produced NFTs in the hippocampus in 10 weeks [150]. This was associated with cognitive impairment, evident as a reduction in novel object recognition compared to controls [151]. This AAV-mediated approach allows for tau overexpression to be induced in adult animals, removing the disruption tau causes during development in constitutive transgenics.

However, the mutations used to induce tauopathies in rodents are predominantly comparable to those seen in frontotemporal dementia (FTD) and parkinsonism, not in AD [152,153]. Indeed, the tau tangles detected in these rodents more closely resemble Pick bodies of FTD than the NFTs of AD [132]. Thus, while tau-only transgenics and seeded models may be useful in understanding the mechanisms by which tau hyperphosphorylation and aggregation spread and cause neuronal loss, they are not reflective of the pathogenesis of AD. One reason for the lack of authenticity of tau models could be that tau hyperphosphorylation and aggregation is not the underlying driver of AD, but rather a consequence of other degenerative processes. We must first identify the key mechanism underlying neurodegeneration in AD before we can reproduce it in rodents.

## 5. Neuroinflammation Hypothesis

There is substantial evidence that amyloid depositions and NFTs trigger inflammatory mechanisms in the CNS [154], including the production of proinflammatory cytokines and the activation of microglia and astrocytes [92,155,156,157]. Yet, there is emerging evidence that inflammation itself may drive the early pathological processes in AD [158,159,160,161]. For example, microglial activation has been reported in patients with mild cognitive impairment in the absence of amyloid pathology [162]. Moreover, individuals with amyloid plaques that lacked evidence of microglial activation were asymptomatic, suggesting that neuroinflammatory responses are required for AD-related cognitive deficits [163]. The discovery of AD risk factors associated with innate immune functions further implicates neuroinflammation as a contributor to sporadic AD [164].

Neuroinflammatory mouse models typically use the intracerebral infusion of chemicals to trigger inflammatory responses and the downstream markers of AD [165,166]. For example, four months of intracerebroventricular infusions of okadaic acid, a toxic inhibitor of serine/threonine phosphatases [167], increased tau hyperphosphorylation, induced Aβ deposition and triggered apoptotic cell death in the rat cortex compared to untreated controls [168,169]. While this model did not recapitulate all the neurochemical markers of AD, for instance there were no NFTs [170], okadaic acid infusion induced significant hippocampal pathology and spatial memory deficits on the Morris water maze [168,169]. 

Inducing neuroinflammation throughout life can also produce AD-like pathology in aging. The polyriboinosinic–polyribocytidylic acid-induced mouse model involves injecting pregnant mice with this inflammatory agent, producing age-dependent increases in tau expression in the offspring. Critically, tau progresses in the spatiotemporal patterns seen in human AD and causes spatial memory impairments in aged animals compared to offspring from untreated mothers [18,171]. Other constitutive changes to neuroinflammation can be produced via the overexpression of cyclin-dependent kinase 5 and p25, which induce tau hyperphosphorylation [172,173], or by knocking out triggering receptor expressed on myeloid cells 2 (TREM2) from APP-mutant mice, which accelerates Aβ pathology and neuronal atrophy [174]. Neuroinflammation may therefore be upstream of amyloid and tau pathology and its models thus better recapitulate the temporal order over which pathology appears in patients. The independence from the overexpression of familial AD genes also suggests neuroinflammatory models may better mimic sporadic AD. Moreover, single nucleotide polymorphisms in neuroinflammatory pathways have also been linked to AD [175] and could therefore inspire future animal models.

Nonetheless, it is difficult to compare findings between constructs as the method to induce pathology varies so greatly. Introduction of inflammatory agents into the brain may not parallel the underlying causes of neuroinflammation in AD, which may arise from a wide variety of insults ranging from vascular abnormalities, mitochondrial dysfunction, to head trauma [164]. Hence, the most fundamental issue with these models is the assumption that inflammation is the first step in the neurodegenerative pathway, rather than a natural response to neuronal injury and damage. The endogenous primary trigger for inflammation in AD remains elusive such that we must resort to injecting pro-inflammatory chemicals into rodents. It is also not clear whether these compounds activate the same inflammatory processes as the human disease. As such, a better model would induce neuroinflammation via the same mechanism as in patients, although we must first understand these processes before such models can be developed. Furthermore, a genetic predisposition to altered neuroinflammation does not circumvent the need for an initial trigger, meaning transgenic models of neuroinflammation are again modelling a response, not a cause, of AD.

In summary, the current approaches to modelling AD in rodents (listed in Table 2) do not faithfully replicate all the features of the disease. They are not based on the underlying mechanisms driving AD, particularly in its initial stages. Consequently, these models do not recapitulate the full neuropathological phenotype and have yielded few effective treatments and no disease-modifying therapies. We therefore need a completely different approach to modelling AD.

## 6. Lessons Learned from Modelling Parkinson’s Disease

A related disorder in which animal models have been used successfully is Parkinson’s disease (PD). PD is a progressive neurodegenerative disease characterised by movement dysfunctions such as bradykinesia, tremor and rigidity [176]. In the later stages of the disease, PD is also associated with dementia and cognitive dysfunction [176]. Only 5–10% of patients have a family history of PD, with just 5% having an autosomal mutation [177]. In these cases, mutations occur in the gene for α-synuclein (SNCA), or in leucine-rich repeat kinase 2 (LRRK2), vacuolar protein sorting ortholog 35 (VPS35) and in genes linked to mitochondrial function [178]. Interestingly, there is a frequently observed co-pathology with AD and sporadic PD [179].

The classical neuropathological hallmark of PD is the presence of cytoplasmic deposits in cell bodies, known as Lewy bodies (LBs), that contain misfolded α-synuclein [180,181]. The progressive loss of dopaminergic neurons in the substantia nigra pars compacta (SN) and the degeneration of its projections to the striatum cause the characteristic motor dysfunction in PD, which correlates with disease severity and latency of disease onset [182,183,184]. When motor symptoms emerge, a significant number of dopaminergic neurons in the SN and terminals in the striatum have been lost [184,185,186,187,188], with up to 90% of dopaminergic neurons having degenerated by the later stages of PD [182,184,189].

Drug-induced parkinsonian movement dysfunction has classically been used to study PD in rodents. The most widely used model has been to lesion dopaminergic neurons with the neurotoxin 6-hydroxydopamine (6-OHDA). 6-OHDA is taken up selectively into catecholaminergic neurons, where it is oxidised to yield hydrogen peroxide and reactive oxygen species, resulting in cell death [190,191]. When injected unilaterally into the midbrain or striatum, 6-OHDA kills dopaminergic neurons and causes rats to rotate towards the contralateral side [192]. These animals also show deficits on the rotarod test, in that rodents with dopamine lesions have a reduced latency to fall off the apparatus compared to controls (e.g., [193]).

An alternative method of inducing dopaminergic neurodegeneration is by infusing the mitochondrial complex I inhibitor 1-methyl-4-phenyl-1,2,3,6-tetrahydropyridine (MPTP) into the rodent brain [11,194]. Dopaminergic cells are particularly sensitive to this toxin due to their high metabolic demand [195]. Immunohistochemical staining showed that chronic MPTP infusion in rats destroys dopaminergic neurons [196] and produces LB-like structures in striatal and nigral neurons [197,198]. This degeneration is associated with bradykinesia and akinesia (see [199]), resembling the movement deficits seen in PD. Neurochemical lesions such as those produced by 6-OHDA and MPTP produce more selective and thorough lesions than historical electrolytic or radiofrequency lesioning of the SN, and thus produce behavioural effects more similar to Parkinsonian symptoms [200].

Recently, transgenic mice have been used to model PD. These include mice overexpressing mutant α-synuclein, which exhibit fibrillar α-synuclein and brain-wide neurodegeneration [132]. Unlike pharmacological models, however, constitutive α-synuclein mice do not show evidence of dopaminergic neuron degeneration (reviewed in [201,202,203,204]), suggesting that they are poor models of PD-like pathology, at least regarding the mechanisms underlying the movement deficits. In contrast, conditional or inducible cell-type specific overexpression of α-synuclein, for example by injecting viral vectors containing wildtype or mutant SNCA into the nigrostriatal pathway, produces robust and extensive degeneration of dopaminergic neurons [205,206].

There are several lessons that can be learned from rodent models of PD for the study of AD. Firstly, targeting dopaminergic neurons, either by the infusion of selective neurotoxins or by the cell-type specific overexpression of mutated α-synuclein, is more effective than constitutively overexpressing genes. The ability to induce neurodegeneration in adulthood, compared to developmental overexpression in transgenic models, also better models the development of PD with age, which is perhaps unsurprising given that most cases of PD, as with AD, do not have a strong genetic basis.

Secondly, these PD models highlight how modelling specific aspects of a disease can be valuable for identifying therapies for specific symptoms, rather than trying to tackle the entire disease.

The 6-OHDA model has been used to demonstrate the efficacy of L-dopa (a dopaminergic precursor) and carbidopa (a peripheral decarboxylase inhibitor that increases the bioavailability of L-dopa) for the amelioration of movement deficits produced by dopaminergic cell loss (reviewed in [207]). This is also a good model for the demonstration of basic pharmacological principles, such as the supersensitivity to dopamine and dopamine receptor agonists that emerges after dopaminergic denervation, itself potentially evidence of the compensatory mechanisms early in PD [208]. Nonetheless, the construct used in these models does not reflect the underlying pathological processes in PD; dopaminergic degeneration, as measured by cell loss and α-synuclein immunoreactivity in post mortem PD brains, does not occur until the middle stages of PD (Braak stage 3; [209]). As such, this model has not yielded compounds that can reverse, or even stabilise, dopaminergic cell loss [132]. 

Clearly, we need models for neurodegenerative diseases that reproduce the pivotal, driving neuropathological mechanisms. Models should be inducible in adulthood and lead to the characteristic progression of neurodegeneration throughout the brain without relying on mutations associated with the familial form of the disease. They should also target the specific nuclei most susceptible to the disease. 

## 7. A Novel Approach to AD Highlighting Primarily Vulnerable Nuclei: The Isodendritic Core

Neurodegeneration in AD begins several decades before cognitive symptoms emerge and appears to precede the accumulation of Aβ [210]. Psychological symptoms such as anxiety, depression, and sleep disturbances, so-called “pre-cognitive” symptoms, are common in the early stages of AD [210,211] and suggest that there may be disruption to subcortical structures related to these dysfunctions, before cognitive deficits emerge. Parallels have been drawn between AD and PD regarding their mechanisms, frequent co-pathology, and the overlap in brain regions implicated in the early pathogenesis. A persuasive hypothesis of AD pathogenesis is that a group of interconnecting subcortical nuclei are the first to degenerate in AD, which initially impacts emotionality and sleep and only in later stages causes cognitive decline [212]. Perhaps these selectively vulnerable nuclei should instead be the focus for animal models of AD.

### 7.1. The Isodendritic Core in AD

The isodendritic core (IC) is an evolutionarily conserved network of nuclei derived from the basal plate of the ectoderm in embryonic development. IC extends from the brainstem to the basal forebrain in the adult mammalian brain [213,214], and is so named because it constitutes a region of overlapping dendritic fields [214]. These nuclei are the principal sites of monoaminergic neurotransmitter synthesis, namely noradrenaline in the locus coeruleus (LC), serotonin in the raphe nuclei, dopamine in the SN and ventral tegmental area, and ACh in the basal forebrain (Figure 1A) [117]. Nuclei in the IC send extensive ascending projections to the cortex [215,216], where monoaminergic transmission modulates homeostatic and behavioural responses such as cognition, the sleep/wake cycle, and mood. Due to their broad modulatory effects, these IC cells have also been dubbed ‘global neurons’ [213].

The IC retains its sensitivity to growth and plasticity factors throughout life (not just during neurodevelopment) and is selectively vulnerable to the early neurodegenerative processes in AD [212,219]. NFTs are found across brainstem nuclei in AD [122,220], including in the LC, SN, dorsal raphe nucleus (DRN) and basal forebrain in post mortem samples of patients with early-stage AD and MCI [221,222,223,224,225,226,227]. Critically, IC pathology appears before that of other AD-associated brain regions; the LC shows evidence of NFTs in the absence of Aβ ten years before the onset of cognitive changes [123,228,229,230,231]. The same is true for the DRN, where lesions were observed in all Braak stage I (BB I) patients, and even in 20% of BB 0 individuals [223,232].

Cell loss also occurs in the IC and increases with disease severity [225,233,234]. As much as 88% of LC neurons were lost in the post mortem brains of individuals with AD compared to healthy controls [229,235,236]. Severe cell loss has also been reported in the SN [237] and basal forebrain [9,10]. Such extensive neurodegeneration has functional implications for monoaminergic neurotransmitters, with studies showing that cortical and limbic levels of dopamine, noradrenaline and serotonin were depleted in AD patients compared to controls [222,238,239]. 

Further supporting the notion that IC disruption is critical in the development of AD, the extent of IC neuron damage and loss correlates with the extent of AD pathology and symptoms. A greater number of NFTs in the LC was associated with lower cognitive (Mini-Mental State Exam, MMSE) scores, indicating more advanced degeneration [224]. DRN pathology and serotonergic denervation of the cortex also correlated with behavioural changes in patients, such as altered emotionality and psychosis [223,240,241], further linking IC damage to the precognitive symptoms of AD.

### 7.2. Evidence of IC Degeneration in Current AD Models

Despite the clear link between IC degeneration and the initial stages of AD, current animal models have not yet attempted to reproduce AD-related phenotypes by targeting these subcortical cell groups. Constitutive transgenics, for example, express human mutations throughout the brain. Similarly, neuroinflammatory models typically infuse compounds into the ventricles. Meanwhile, when seeded models introduce Aβ or tau into specific brain regions, it is generally into the hippocampus and cortex, rather than into the IC. Consequently, pathology in these models is initiated in a structure that does not exhibit neurodegeneration until later stages of AD [123] and so fails to replicate the characteristic spatiotemporal patterns of pathology that occur clinically. In APP transgenic mice, for example, there was no evidence of a reduction in the number of cholinergic neurons in the basal forebrain even at 24 months of age [242,243,244,245,246,247].

Established AD models can lead, almost as a side effect, to limited neurodegeneration to the IC. By chance, there is some evidence of age-dependent neurodegeneration of LC noradrenergic neurons in 3xTg-AD and APP/PS1 mice, and in TgF344-AD rats [92,248,249]. Reductions in cortical and hippocampal catecholamine concentrations have also been noted [250,251,252,253]. Nonetheless, these effects have not been detected in all strains [254,255], and evidence suggests that changes in terminal neurotransmitter levels may be driven by local degeneration rather than cell body loss [243,246,256,257,258]. Accordingly, these experimental interventions do not really model AD neuropathology in the IC.

Most studies neglect to investigate possible pathology in the brainstem [259,260], and this holds true even in approaches that claim to fully phenotype a strain: for example, a recent behavioural and neurochemical study of 5xFAD mice did not measure amyloid or tau pathology in the midbrain or IC cell loss [261]. Similarly, studies of seeded models focus on pathology in the treated brain regions and subsequently in cortical projections, rather than in the IC [108,146,147]. The lack of evidence in this field should not be confused with a lack of importance; instead, it serves to highlight the need for a new animal model that targets the IC. 

Of particular importance is that the current animal models poorly replicate the precognitive symptoms of AD (for detailed review, see [262]) that can start 10 to 20 years before cognitive symptoms and are a central feature of the early disease stages. Taking the emotional disorder of anxiety as an example, some transgenic mice exhibit increased anxiety on tests such as the elevated plus maze and light dark box (e.g., 3xTg-AD [263], Tg2576 [264]), and increased anxiety has been detected in aged TgF344-AD rats [92,94]. Yet, there is a lot of variability between models; many studies report no change in anxiety-like behaviour [252,265,266], and others see a reduction [267,268,269]. Opposing changes are often seen in males and females of the same strain [270] and altered anxiety-like behaviour tends to emerge with age [271]. Some mice strains do see changes to sleep/wake behaviour—for example, altered sleep bout durations and disruption to circadian rhythms [272,273] that are evident in young mice [274]. Yet, it is unclear whether these changes precede cognitive deficits or are present concurrently. Albeit-limited evidence therefore suggests that transgenic mice do not exhibit the behavioural changes associated with IC degeneration that occur in AD. We therefore need a model that exhibits these early behavioural changes, not just deficits in memory and cognition, to identify prodromal biomarkers that may be useful for the development of preventative therapies.

### 7.3. A Novel Approach to Modelling the IC in AD

To replicate AD-like neurodegeneration, novel animal models should aim to induce dysfunction in the IC. Mechanical or electrolytic lesions, for example, may be able to destroy individual brainstem nuclei; however, these neurons mediate essential basic functions, such as thermoregulation, breathing and key survival behaviours, such that their destruction would result in the death of the animal. Chemical lesions could be used to target specific neurotransmitters, an approach that resembles the dopamine depletion studies in PD. Yet, the IC consists of heterogeneous transmitter systems of dopamine, noradrenaline, acetylcholine, and serotonin. Targeting just one of these neurotransmitter systems, as demonstrated with the lesions used to investigate the cholinergic hypothesis, will not recapitulate the entire disease or its progression. We would therefore need to identify a bioactive agent common to all nuclei, irrespective of transmitter.

Indicative of their shared evolutionary origin, neurons in the IC express the enzyme AChE, irrespective of their primary transmitter: hence, this familiar protein acts also in an unfamiliar capacity unrelated to is catalytic action [275,276]. The salient part of AChE underlying its bioactivity in relation to AD is a 14mer peptide cleaved from its C-terminus, ‘T14’, generated from exon 6 of the AChE gene (Figure 1B). T14 is proposed to cause AD by recapitulating developmental mechanisms in the ageing brain, where events such as large increases in intracellular calcium or neuronal outgrowth are excitotoxic and thus cause cell death [218,277]. T14 is elevated in the human midbrain as early as the presymptomatic Braak stage II [217,218,278], and in vitro studies have shown it to induce excitotoxicity and reduce cell viability [279]. Critically, T14 appears to operate upstream of Aβ and tau in that it triggers production of both these markers [280]. In fact, pharmacologically blocking T14 binding to its receptor, an allosteric site on the α7 nicotinic receptor [279], reduced Aβ and tau load in the brains of 5xFAD mice, suggesting T14 is vital in their accumulation [217].

Manipulation of IC function via the upregulation of the T14 system, for example by directly injecting the peptide or viral overexpression of T14 in AChE-expressing neurons in the rodent basal forebrain (analogous to the NbM in humans [281]) or LC, is therefore proposed as a novel model of AD (Figure 1C). This model would circumvent the previous difficulties posed by uncertainty of both anatomical and chemical targets in other models by directly targeting the cells proposed to underly AD, not those that simply degenerate with its progression, like the hippocampus. In this case, it would be inappropriate to simply target cholinergic transmission systemically (e.g., pharmacologically), as this would not differentiate between the vulnerable midbrain AChE-containing neurons (not all of which release ACh) and cholinergic interneurons and lower motor neurons that do not readily degenerate in AD [31]. Cholinergic pharmacology would also not affect other monoaminergic transmitters such as noradrenaline and serotonin, whereas T14 is posited to be equally efficacious across neuronal subtypes.

Consequently, we propose that an increase in T14 levels would trigger Aβ accumulation and tau hyperphosphorylation, ultimately leading to cell death and cognitive deficits in these animals (Figure 1D). Given T14 levels start to rise before Aβ, it may be possible for this model to recapitulate the early symptoms of AD, including anxiety and sleep disturbances. This model could also be used for drug discovery and lead to the identification of related novel, potentially druggable targets, which would not be possible without such an authentic model. Of course, a T14 model would still have its limitations—for example, it would not address the range of initial insults that could trigger the accumulation of T14, namely trauma (such as head injury or ischemia [218]). Moreover, we do not yet know how much T14 is generated by such an event, nor the timeframe in which it accrues, to reproduce a quantitative parallel in rodents. Nonetheless, this novel construct would be a major advance towards modelling an upstream cause of AD without the need for FAD-associated mutations.

## 8. Conclusions

For decades, researchers have attempted to model AD by recapitulating downstream markers of its pathology, including cholinergic neuron loss, Aβ and tau aggregation, and neuroinflammation, rather than exploring its true underlying mechanism. Insights gained from the study of PD show us that we must model the actual neurodegenerative mechanisms of AD in adult, wildtype animals by specifically targeting the neural populations first affected by a disease. A model, by definition, should capture the salient features of its subject, at the expense of the extraneous ones: in AD, anatomy and neurochemistry both need to be considered as salient factors. We propose an alternative model, based on the aberrant accumulation of the novel peptide T14: we suggest that this novel preparation would produce a more authentic and fuller profile of AD progression. Only once we understand the initial causes of the disease can we hope to fully understand progression, identify biomarkers, and ultimately find an effective treatment for AD. 

## Figures and Tables

**Figure 1 ijms-25-06222-f001:**
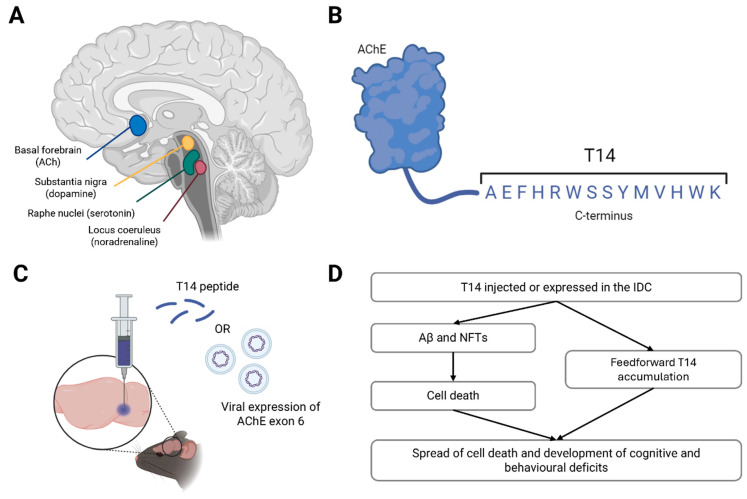
A novel animal model of AD. Schematic of the isodendritic core (IC) in the human brain, highlighting the main cholinergic, serotonergic, dopaminergic and noradrenergic nuclei (**A**). The structure and sequence of T14 at the C-terminus of acetylcholinesterase (AChE) (**B**). Proposed mechanisms of delivery of T14 to the IC in laboratory rodents, either via direct infusion of the peptide or the viral overexpression of exon 6 of AChE (which encodes T14), into the target nuclei (**C**). Schematic of the hypothesised effects of T14 administration into the rodent brain, including amyloid beta (Aβ) and neurofibrillary tau tangles (NFTs), mimicking AD progression, based on data published in [217,218] (**D**). Figures made using BioRender.

**Table 2 ijms-25-06222-t002:** Summary of approaches to induce AD-like pathology in rodents.

Feature	Transgenic Models	Seeded Models	Inflammatory Models
Causal agent	Overexpression of one or more human AD-associated mutations to APP, PS1 and/or MAPT	Surgical injection of Aβ, tau or their associated aggregates into the brain, or AVV-viral transfection of tau mutants	Infusion of pro-inflammatory chemicals or overexpression of inflammatory markers
Location	Brain-wide, varied distribution based on promoter used to express transgene	Targeted to hippocampus or cortex	Brain-wide, intracerebroventricular infusion or constitutive transgene expression
Species used	Predominantly mice	Mice and rats	Mice and rats
Age at induction	Expression throughout life	Adulthood	Adulthood or expression throughout life
Sex of animals	Typically female as pathology is more robust	Both	Both

## Data Availability

No new data were created or analysed in this study. Data sharing is not applicable to this article.

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
