# Peer review of "Rodent Models of Alzheimer’s Disease: Past Misconceptions and Future Prospects"

_ijms, 2024, doi:10.3390/ijms25116222_

Round 1

Reviewer 1 Report

Comments and Suggestions for Authors

The review article is interesting and I have read it with curiosity, due to my research interest in neurodegeneration. The manuscript is well written and it is easy to follow the discussion. However some improvement in the style and organization should be perfoemd. I commented on the possibility to improve the paper because of in my opinion some aspects should be better treated. Below my comments:

-I recommend to improve the last part on the unique novel method to model the AD in animals. after the description (maybe a figure should help to understand the procedure to obtain the AD model), I ask if the model was already used for the possible drug discovery approach, also considering related novel, target that could arise from a reliable model with respect to the current ones.

-Also for the previous part maybe picture(s) on the different models of AD can improve the quality of the paper

- the conclusion should be enriched with a personal point of view and how based on the experience of authors will be possible to obtain a reliable models, speculating on this issue, adding a critic discussion on that. The limitation part of the current model is well highlighted, while in my opinion some info on the limitation on the presented novel model should be clearly stated as well as  possible approaches for overcoming the limitations

-in the introduction, it is nice that the authors briefly consider the current approach to treat AD, including the recent monoclonal antibodies Aducanumab and the related one approved by FDA, but not by EMA, discussing also in this case the difficult in this sense as marginally outlined.

-row 77 authors wrote ...human polymorphisms in the apolipoprotein E (APOE)...in particular the genotype  is APOE epsilon4, please add the correct information. The same at row 92... overexpressing human APOE what kind of APOE?

-row 135  amyloid levels should be  Aβ ? the same row 142. Authors should pay attention to the acronym used. please check throughout the text.the same row 229 only for citing some other discrepancies between acronyms and the text

-Table 2 authors introduced only a ref in the last row, Is there a need to insert other refs for the rows in which the refs were not present?

-row 218 tau tangles authors used the acronym NFTs elewhere, please uniform the wording

Comments on the Quality of English Language

minor

Reviewer 2 Report

Comments and Suggestions for Authors

In their review, Collins and Greenfield are trying to attract researchers’ interest to animal models of Alzheimer’s disease (AD). The review is timely, since AD research needs to make some radical turns and start searching for new directions to find a meaningful way forward. Novel approaches and new animal models are very much needed, and the existing ones must be better understood and used in more judicious manner. New animal models could break the impasse and in this sense authors’ manuscript represents a praiseworthy effort and distinguishes it from the multitude of AD reviews appearing all over the place all the time . Going through the text, I am findings occasional shortcomings, question marks and possible gaps which I would like authors to address and answer.

I summarize my comments below:

  # 1    Line 59: You might consider mentioning some of these AChE inhibitors specifically (tacrine?); it was fascinating to watch in 1990’s how, after the fall of the iron curtain, laboratories popped up on both sides of the former divide with almost ready-made AD therapeutics inspired by previous defense research. You could return to tacrine later (# 14) when discussing the cholinergic neurons of the isodendritic core and state more explicitly why tacrine would be unsuitable/less suitable in the approach you are suggesting.

  # 2    Line 61: One lesson from PD research (see later in the text) it is that a very high proportion (possibly > 75%) of dopaminergic neurons in SNpc has to go before we see any signs of PD. What percentage of neurons has to disappear in AD before we can identify onset of dementia? Any data? From imaging?

  # 3    Line 64: I have heard from neuroanatomists that the basal forebrain in rats is not exactly the same thing as the nucleus basalis (of Meynert) in humans. Can you do some searching around to clarify this question? It could be of crucial importance in the proposed modelling.

  # 4    Line 125: It might help to clarify what exactly was deposited (or transferred?). Otherwise, less enlightened reader  might think of Abeta42 administration or some such thing.

  # 5    Lines 133-135. The statements here apparently clash with # 4. You may chose to restate it more explicitly and specify what exactly happens or does not happen following the injection; I found it a bit confusing.

  # 6     Line 143: “tau tangles” before amyloid deposits – wouldn’t you think that this is evidence that AD originates pre-synaptically (if it originates in neurons at all and not, say, for the sake of argument, in microglia or blood vessels)?

  # 7     Line 156: “fail in clinical trials” This is indeed curious and has been noticed by lay public, too: https://www.statnews.com/2019/06/25/alzheimers-cabal-thwarted-progress-toward-cure/ I should add that Sharon Begley cannot be easily dismissed; she was highly regarded as a science journalist. The failure of amyloid-target drugs and the generally dismissive tenor of your discussion of amyloid models might need more detailed explanation and further buttressing by additional references. It could be very important in the context.

 # 8      Line 188: The loss of short-term memory early in AD could indeed correlate with early changes in hippocampus (or even amygdala?), I should think. If there is specific evidence of the correlation, perhaps from imaging studies, you may wish to include it and be more explicit, to make for smoother/easier reading.

 # 9      Line 196: Again, you may wish to add what exactly was seeded – tangles, pathological tau, hyperphosphorylated tau?

 # 10   Line 252: I agree that there is a multitude of potential causes of neuroinflammation some of which might perhaps be relevant in AD. Those mentioned earlier in the paragraph (head trauma, vascular etc.) indeed do not easily lend themselves to replication in animals as experimental models; yet there have been mutations (SNP’s) recently linked to AD, potentially promoting inflammation (eg. in CD36); these could provide inspiration for future animal models.

 # 11   Line 293: One lesson earned from the PD research is that there has to be a massive loss of dopaminergic neurons in SNpc (which itself contains about 80% of all DA neurons in the human brain). I have seen many elderly  brainstem postmortem with clearly reduced DA neurons in SNpc, sometimes even asymmetrically (early PD is very often asymmetrical), yet no mention of PD in antemortem pathology. How much neurodegeneration would you have to detect in (a) hypothetical nucleus/ei to link it specifically to AD pathology?

 # 12   Line 387: I seem to remember that early experiments with mechanical lesions in SNpc were not very successful in producing PD signs. Using neurotoxins such as 6-hydroxydopamine was more successful (MPTP+ perhaps too much).

 # 13   The IC hypothesis is inherently attractive since many early signs and symptoms of dementia could be explained by changes/neurodegenerations somewhere within the region (sleep disturbances, wild dreams, mood changes etc.). The question is where to start – let’s not forget that IC is roughly coextensive with what used to be called reticular formation; despite major progress over the past forty years or so it is still a bit of a jungle – where do you want to start? Can you suggest/list toxins specific for any of the neurotransmitters (biogenic amines, neuropeptides) typical of IC?

 # 14   Line 397: What makes you think that manipulating T14 would be more successful than tacrine et al.? How would it “ ..circumvent the previous difficulties.. “ (lines 400/401). How would you distinguish, in the T14 manipulation, the IC cholinergic neurons and distinguish them from say, lower motor neurons or preganglionic autonomics?

Minor comment/query: Are there any “spontaneous” animal models, ie has an old age dementia been detected in animals such as rats, cats, guinea pigs etc which could be useful in research? I have heard of AD in mountain gorillas in the wild and other primates (gorillas, chimps) in captivity but, presumably using them for research might be neither ethical nor useful. Wouldn’t it be odd if AD turns out to be restricted to primates? What practical consequence would it have for rodent animal models?

Round 2

Reviewer 1 Report

Comments and Suggestions for Authors

I wish to thank the authors for taking into account my comments. The manuscript has been significantly improved, and I recommend the acceptance of the revised version.

Comments on the Quality of English Language

minor editing is requested

Reviewer 2 Report

Comments and Suggestions for Authors

Authors addressed all my queries and made changes/amendments to their MS, where appropriate.

I especially like the emphasis on the poverty of amyloid hypothesis, incl. in Abstract.

Minor query: wouldn't you say "..tacrin in 1989.." in (better) English? If so, fix in the proofs, please.